# Hepatitis E Virus Research in Brazil: Looking Back and Forwards

**DOI:** 10.3390/v15020548

**Published:** 2023-02-16

**Authors:** Jaqueline Mendes de Oliveira, Debora Regina Lopes dos Santos, Marcelo Alves Pinto

**Affiliations:** 1Laboratório de Desenvolvimento Tecnológico em Virologia, Instituto Oswaldo Cruz, Fiocruz, Rio de Janeiro 21040-360, Brazil; 2Departmento de Microbiologia e Imunologia, Universidade Federal Rural do Rio de Janeiro, Seropédica 23890-000, Brazil

**Keywords:** hepatitis E virus, seroprevalence, animal hosts, pathogenesis, non-human primates

## Abstract

Hepatitis E virus (HEV) has emerged as a public health concern in Brazil. From the first identification and characterization of porcine and human HEV-3 strains in the 2000s, new HEV subtypes have been identified from animal, human, and environmental isolates. As new potential animal reservoirs have emerged, there is a need to compile evidence on the zoonotic dissemination of the virus in animal hosts and the environment. The increasing amount of seroprevalence data on sampled and randomly selected populations must be systematically retrieved, interpreted, and considered under the One Health concept. This review focused on HEV seroprevalence data in distinct animal reservoirs and human populations reported in the last two decades. Furthermore, the expertise with experimental infection models using non-human primates may provide new insights into HEV pathogenesis, prevention, and environmental surveillance.

## 1. Introduction

Hepatitis E virus (HEV) is a quasi-enveloped or non-enveloped, positive-sense single-stranded RNA virus comprising 6.8 to 7.2 kb in length, organized in three open reading frames (ORFs). HEV strains are classified in the Family *Hepeviridae* into two subfamilies that infect mammals and birds (*Orthohepevirinae*) or fish (*Parahepevirinae*), five genera (*Paslahepevirus*, *Rocahepevirus*, *Avihevirus*, *Chirohepevirus*, and *Piscihepevirus*), and ten species. The species of zoonotic origin, *Paslahepevirus balayani* and *Rocahepevirus ratti*, include HEV strains derived from humans and several mammalian species [1,2,3]. Members of the *Paslahepevirus balayani* species are classified into eight genotypes and several subtypes. The updated list of proposed reference sequences for *P. balayani* subtypes classification now includes the subtypes 1g, 3k, 3l, 3m, and 8a [4]. Genotypes 1 and 2 (HEV-1 and HEV-2) exclusively infect humans, whereas genotypes 3 and 4 (HEV-3 and HEV-4) infect humans and other animal species, mostly swine; genotypes 5 and 6 have only been found in wild boars, whereas genotypes 7 and 8 were isolated from camels, with one report of HEV-7 transmission to human - a liver transplant recipient who regularly consumed camel meat and milk [5]. Of note, the genus *Rocahepevirus*, which infects rodents, shrews and carnivores, can also infect humans [6,7].

The virus is primarily transmitted via the fecal-oral route from contaminated water or food. The World Health Organization (WHO) estimates that HEV infection affects about 20 million people in developed and developing countries, causing 44,000 deaths a year due to acute severe or fulminant hepatitis [8]. HEV is prevalent worldwide, but the disease is most common in East and South Asia, and Africa. In Brazil, as in other countries in South America, HEV-3 is, currently, the only genotype identified in humans [9,10,11,12], animal hosts [13,14,15,16,17,18,19,20,21,22,23,24], and environmental matrices [14,25,26].

Hepatitis E has emerged as a public health concern in Brazil since the HEV-3 subtypes identified in domestic pigs are closely related to human HEV-3 strains and, therefore, must be considered under the One Health concept. In this review, we primarily focused on the species *Paslahepevirus. balayani*, which is presently responsible for most HEV infections in humans. Here, we review and discuss Brazilian studies from the serological and molecular identification of HEV-3 in swine and humans; HEV seroprevalence data in distinct human populations; experimental infection studies using animal models; and perspectives on prevention and environmental surveillance.

## 2. Hepatitis E Virus in Animal Reservoirs in Brazil

Except for the *P. balayani* genotypes 1 and 2, it is known that HEV, among other relevant viruses that cause hepatitis, is well characterized as a zoonotic virus. After the identification of a swine HEV-4 in the midwestern United States [27], worldwide research groups have developed serological and molecular investigations in several other animal hosts, including wild boars [28], deers [29], wild rats [30,31,32], chickens [33], fish (cutthroat trout) [34], rabbits [35], camels [36], and bats [37]. Swine (domestic pigs and wild boars) and cervids are considered potential reservoirs due to their epidemiological importance [38,39].

In Brazil, the first study searching for serological evidence of HEV infection in animal reservoirs was developed by our research group in 2004–2005, using a standardized enzyme immunoassay [40] modified to detect species-specific anti-HEV IgG antibodies in several animal species [41]. Anti-HEV IgG, specifically for the respective species, were detected in cows (1.4%), dogs (7%), chickens (20%), and pigs (24.3%) from ten commercial farms in Rio de Janeiro state. Additionally, wild rodents (*Nectomys* sp.), which were randomly captured within a commercial swine farm, also had anti-HEV IgG detected. However, HEV RNA and genotypes were not investigated in this study [41]. Similarly, Kabrane-Lazizi et al. [30] and Favorov et al. [31] found a high anti-HEV positivity (59.7% and 44%-90%, respectively) in wild rats from the United States. A further study reported the isolation of the zoonotic HEV-3 (genus *Paslahepevirus*) from the liver tissues of one of the 446 rats examined [42]. Furthermore, rat HEV-C1 (genus *Rocahepevirus*), which can also cause zoonotic infection, has been detected in wild rats in many countries in Europe, Asia, and North America [43]. In Brazil, novel *Rocahepevirus* species (formerly *Orthohepevirus* species C) were identified in the murine rodents *Necromys lasiurus* (1.19%) and *Calomys tener* (3.66%) [44].

From the serological evidence of HEV circulation in several animal species in Brazil, a subsequent study showed the characterization of HEV-3 in swine herds from Rio de Janeiro and Mato Grosso states (southeast and midwest regions, respectively), being the first description of this genotype circulation in the country [13]. This research group also showed the detection of HEV-3 in swine and effluent samples from slaughterhouses [14] and the first autochthonous human case [9], raising the discussion of zoonotic transmission potential. Afterwards, several studies debated the zoonotic aspect of hepatitis E in Brazil. Studies developed in the south region showed the circulation of HEV subtype 3b in fecal, liver, and bile samples from healthy pigs in different growing stages. Results revealed a potential source for HEV infection for consumers of pig livers or workers directly dealing with animals or products of animal origin [15,16]. An investigation developed in the Eastern Brazilian Amazon showed the co-circulation of subtypes 3c and 3f in slaughtered swine from different regions of the Pará state, pointing to a genetic potential for recombination and the emergence of new subtypes [17]. In 2013, another study developed in Mato Grosso state evaluated HEV circulation in pigs from large-scale and family-scale farms. The presence of HEV antigens in fragments of the small intestine sample, shown by immunohistochemistry, evidenced the risk of zoonotic transmission. The study showed the co-circulation of subtypes 3b and 3f, adding more information on HEV genetic diversity in the country [18].

Further investigations showed high seroprevalence and HEV-3 detection in swine and pork products from the South [21,22,25,26], Northeast [23,24], and Midwest regions [19]. In 2019, HEV-3 from swine in the Northeast region was partially sequenced, and the study revealed the proximity of HEV subtype 3f to human isolates from Brazil. Notably, parsimony ancestral states analysis indicated gene flow events from HEV cross-species infection, suggesting a significant role of pig hosts in viral spillover (24). Thus far, all studies that detected HEV in swine populations in Brazil have shown a close relationship with human isolates. These data suggest that HEV’s zoonotic ability, not to mention the possibility of adaptation to new animal species, must be considered an important issue for human and animal health.

Considering other potential reservoirs, de Souza and colleagues, in 2018, described a study in rodents from São Paulo State. In this study, the authors applied a high-throughput approach and identified novel orthohepeviruses in blood samples from *Necromys lasiurus* and *Calomys tener* species (*Cricetidae*), bringing perspective to the host range and viral diversity of the *Hepeviridae* family in Brazil, including the genus *Rocahepevirus* [44]. Interestingly, a rodent HEV strain, which was further detected in Hungary, showed close phylogenetic relation with the Brazilian strain detected in *Necromys Lasiurus*, but clearly separated from other *Muridae*-associated strains, suggesting the presence of a *Cricetidae*-specific genotype in Europe and South America [45]. Recently, Severo and colleagues also described a seroprevalence of 13.1% among wild boars hunted for population control in Santa Catarina (SC) state, south of Brazil. The study aimed to evaluate the health status suggesting a potential risk to public health and swine herds, considering the One Health approach [46]. Original data from another study developed in the South region of Brazil showed the detection of HEV-3 in bovine liver samples commercialized in butcheries [47]. Thus far, other than in swine and wild boars, HEV-3 (subtype 3f) was recently detected in one capybara (*Hydrochoerus hydrochaeris*), a large rodent found in many urban parks in Brazil [48]. The data presented above reinforces the necessity for studies to discuss inspection regulations and food security issues.

## 3. HEV Prevalence in the Brazilian Population

HEV is primarily transmitted via the fecal-oral route, being the principal etiological agent of acute viral hepatitis in Asia and Africa, where it has been associated with waterborne outbreaks. Therefore, from 1990 to 2000, HEV prevalence studies focused on impoverished communities settled in rural and urban areas at the neighborhood of big cities [49,50] or living in precarious sanitary conditions in remote Brazilian Amazon villages [51,52,53,54]. In Rio de Janeiro, Trinta et al. [49] assessed distinct (presumptive) risk groups, such as asymptomatic individuals living in urban and rural areas, blood donors, intravenous drug users (IVDU), hemodialysis patients, and patients with acute non-A, non-B, and non-C (non-A-C) viral hepatitis referred to the Laboratory of Viral Hepatitis, IOC/Fiocruz, from 1994 to 1996 [49]. Similarly, Paraná et al. assessed patients from a referral hospital for liver diseases and blood donors from Salvador, Bahia state [55] (Table 1).

### 3.1. Acute Non-A, Non-B, and Non-C Hepatitis Patients

The earliest studies in Brazil reported an alleged outbreak and high HEV prevalence rates in patients with acute hepatitis from Brazilian Amazon villages (12%) and urban areas from the Northeastern region (17% to 29%) [55]. In contrast, Trinta et al. [49] found a low HEV seroprevalence in acute non-A-C hepatitis patients (2.1%) from Rio de Janeiro (Southeastern region), even lower than in blood donors (4.3%) and equivalent to that found in inhabitants from rural (2.1%) (49) and urban areas (2.1%) [50] from the same city (Table 1). Important to note that two anti-HEV EIAs, the commercially available enzyme immune assay (EIA) Abbott Laboratory (Chicago, IL, USA), and a new EIA using two recombinant proteins developed and standardized by Obriadina et al. [40,49], were used in Rio de Janeiro seroprevalence studies. Such a discrepancy in anti-HEV seropositivity (2% to 38%) between the studies developed in Northeast and Southeast regions cannot be only attributed to socioeconomic/regional differences. Indeed, further studies employing a different EIA reported a lower anti-HEV prevalence (5.3% to 5.9%) in acute non-A-C hepatitis serum samples retrospectively analyzed [65,66] (Table 1). Most likely, it could have been due to the lack of sensitivity/specificity of the first-generation anti-HEV IgG/IgM commercialized EIA used in the earliest serological surveys (1990–2000) (Table 1).

The first autochthonous human case of acute hepatitis E in Brazil was retrospectively identified and molecularly characterized by Lopes dos Santos and colleagues from the Viral Hepatitis and Technological Development in Virology laboratories, IOC/Fiocruz [9]. The 30-year-old patient reported a history of undercooked pork meat consumed a few weeks before the onset of acute hepatitis. This HEV strain clustered within HEV-3 (subtype 3b) sequences and was closely related to porcine HEV-3 strains previously identified in Rio de Janeiro [13]. These findings evidenced the cross-species infection with HEV-3, probably via the food-borne transmission route, with implications for public health. Two further studies retrospectively evaluated HEV RNA and antibodies prevalence in serum samples from sporadic cases of acute non-A-C hepatitis patients. Although anti-HEV IgM have been detected (0.3% and 3.4%) in both cohorts, HEV RNA was undetectable. [65,66]. Thus far, few sporadic cases but no outbreaks of hepatitis E have been notified to the Brazilian Ministry of Health.

### 3.2. Rural Populations

The broad circulation of the zoonotic HEV-3 among Brazilian swine farms motivated several research groups to investigate HEV prevalence in individuals at potential risk of zoonotic or occupational transmission. Vitral et al., 2005 reported a 6.3% anti-HEV prevalence among swine handlers from Rio de Janeiro state [41]. A further cross-sectional study developed in several swine farms in Mato Grosso state (Central Brazil) found an 8.4% (95% CI, 5.6% to 12.2%) anti-HEV prevalence among individuals exposed to swine [56] (Table 1). Most of them lived on subsistence family farms, without access to potable water or public sewage systems. The control group, composed of blood donors from the urban area who had never handled swine, showed a lower (4%, 95% CI, 1.3% to 10.4%) anti-HEV prevalence (Table 1). However, the multivariate analysis did not confirm the association between anti-HEV positivity and the occupational risk of infection.

Further studies focused on remote areas of the Brazilian Amazon Basin - mainly Acre and Rondonia states, which have been destinations of increasing floods of migrants from central-south Brazil, since the 1970’s. The National Integration Program, a colonization project originally designed to resettle 100,000 families between 1971–1974, reached a million families by the 1980’s. Intermittently, migration “booms” to Amazonian states have occurred whenever there is a large demand for a frontier product or natural resource stimulated massive rural-urban migration to the region. However, for small scale agriculturists, such overcrowded settlements do not provide an alternative to poverty [86]. Consequently, poor housing conditions, disorganized human concentrations, a lack of sanitary infrastructure, and predatory invasion of the forest determine the outbreak of important malaria outbreaks, with high levels of morbidity and associated mortality [87].

Aiming to evaluate the prevalence and risk factors for HEV infection between people living in agricultural settlements, and in urban, rural, or riverine communities of the Brazilian Amazon Basin, a total of 1,831 serum samples obtained in cross-sectional studies—designed to evaluate the epidemiology of malaria and other intestinal parasites—between 2004 and 2013, were retrospectively analyzed. Anti-HEV IgG/IgM antibodies were detected by EIA using the Biokit (Barcelona, Spain) or Mikrogen (Neuried, Germany) kits commercially available at the time of the retrospective serological analysis (2013–2016). In one of the largest agricultural settlements in Acre, the westernmost state of Brazil, located in the southwestern Amazon, Vitral et al. [57] developed a cross-sectional study to evaluate and compare the prevalence and risk factors for HAV and HEV infections in this subset. Of interest, the authors did not observe a significant spatial clustering of HAV and HEV seropositivity, despite the high HAV seroprevalence (82.9%) found in this population. Although anti-HEV prevalence (12.9%; 95% CI, 9.5% to 16.2%) was much lower than that observed for HAV, Acre’s agricultural settlers presented a considerably higher HEV prevalence compared to those reported in rural settlements of Central Brazil [(3.9%; 95% CI, 2.8% to 5.4%), and (3.3%; 95% CI, 1.6% to 6.1%)] [59,60] (Table 1). Interestingly, a similar anti-HEV prevalence rate (14.6%) was found in rural settlers from Rondônia state, which is also located in the south-west Amazon [58] (Table 1). Both Rondônia and Acre states present very low human development index (0.66–0.69), with approximately 70% of their populations living in rural areas [88]. It is essential to consider that Acre and Rondônia’s agricultural settlements are mainly composed of migrants from the southern and southeastern regions and, indeed, are not representative of the native inhabitants. Interestingly, the southern and southeastern regions present the highest anti-HEV prevalence rates in Brazil regarding seroprevalence studies in general populations and blood donors (Table 1).

### 3.3. Brazilian Traditional Peoples

Studies aiming to determine the epidemiological features of HEV infection in Brazil have assessed HEV seroprevalence among “Povos e Comunidades Tradicionais” (Traditional Peoples and Communities) [89]. According to the National Policy for Sustainable Development, traditional peoples in Brazil are “culturally differentiated population groups that recognize themselves as such. They occupy and use territories and natural resources as a condition for their cultural, social, religious, ancestral, and economic organization, using knowledge, innovations and practices generated and transmitted by tradition”. This concept interweaves indigenous, afro-descendant (“quilombolas”), and riverine. Brazilian riverine people can be descendants of Africans, Europeans, or from Brazilian native peoples. They primarily live in wooden houses built on stilts extending across river and stream banks; therefore, riverine communities are characterized by their floodplain livelihood [90].

The high endemicity of HEV infection in developing countries directly correlates with the lack of potable water and sanitation access. For instance, in South-East Asia, the use of river water for drinking and cooking is significantly associated with HEV genotype 1-associated hepatitis. Therefore, the anti-HEV prevalence in the riverine population is usually very high (40% to 60%) [91]. In contrast, a low HEV seroprevalence (approximately 4%) was found in Brazilian riverine communities from the Western Amazon [54] and Eastern Amazon [61]. The former study investigated the prevalence and risk factors for viral hepatitis in riverine families from the Furo do Maracujá island, located in the Pará state, North Brazil (Table 1). This insular population, whose main economic activity is the cultivation and extraction of a berry fruit called açaí (Euterpe oleracea Mart.), was selected for sampling due to its remote location (1° 22’ 23.502” S, 47° 52’ 20.593” W) and poor sanitation conditions [90]. The low HEV seroprevalence contrasts with the very high HAV endemicity in those regions. Although water samples have not been assessed, there have been no report of outbreaks and sporadic cases of acute HEV infection associated with HEV genotype 1 (which is prevalent in East and South Asia, and Africa) in Brazil so far.

In Brazil, an estimated three thousand afro-descendent communities — the so-called “quilombolas” — are distributed throughout the five geographic regions. Historically, Quilombos were established in Brazil during the colonization period (1500–1822), which was marked by the exploitation and enslavement of native (indigenous) and Afro-descendant peoples [92]. Because the Quilombolas communities are predominantly located in isolated or semi-isolated rural areas, Souza et al. [62] realized a study aiming to describe the demographic and epidemiological characteristics of afro-descendant rural communities from Pará state, Eastern Brazilian Amazon. The studied population settled near the forest, under precarious sanitation conditions and without access to safe water. Notwithstanding, the study disclosed a very low anti-HEV IgM/IgG seroprevalence (1.6%), lower than those reported in riverine communities (4.0%) [54,93], and rural settlers (12.9% to 14.6%) from the same region [57,58] (Table 1).

HEV prevalence data in Brazilian indigenous communities are scarce. Villar et al. [63] recently reported a very low (0.19%) anti-HEV prevalence in the Apinajé indigenous community in Brazilian western Amazon. Preliminary results of a cross-sectional developed in indigenous communities living in the Brazilian Amazon Rainforest showed a low anti-HEV prevalence (2.8% to 5.7%) (Table 1). This study was designed to evaluate the prevalence of malaria and intestinal parasites coinfection in the Yanomami Marari community [64]. Malaria diagnoses were carried out at the Laboratory of Immunoparasitology, Fiocruz, Rio de Janeiro. Parasitological analyses were carried out at the Tropical Medicine Centre, Rondônia. Anti-HEV analyses were carried out at the Laboratory of Technological Development in Virology—Fiocruz, Rio de Janeiro.

### 3.4. Occupational or Behavioral Risk Groups

Further studies have focused on individuals at (presumptive) high risk of exposure to HEV, such as urban garbage collectors, and illicit non-injecting drug users. Recyclable waste pickers (RWP) have close contact with garbage, wasted food, and polluted water, being at high occupational risk of infection with gastroenteric pathogens [94]. Although hepatitis E and A viruses can be transmitted by contaminated water or food, the occupational risk for those workers has been less assessed. In a cross-sectional study enrolling people working at waste recycling cooperatives, the authors estimated a 5.3% (95% CI, 3.7% to 8.3%) IgG anti-HEV prevalence (Table 1), which does not differ from overall estimates for the general Brazilian population [73]. Similarly, a case-control study was developed in Mexico to evaluate the occupational risk of HEV infection [95]. The authors reported a higher HEV seroprevalence (16.3%) in waste pickers than in the control group (9.3%). Further studies are needed to confirm whether working on garbage collection sets represents a risk of HEV infection.

In Brazil, crack-cocaine abuse, a drug scene mostly related to vulnerability and social exclusion, has emerged as a serious public health issue. The potential risk of HEV infection associated with crack-cocaine (CCU) has been evaluated in recent cross-sectional studies, which showed high anti-HEV prevalence rates (14.2%, 18.1%, and 20%) in sampled populations from Midwest, North, and South regions [70,74,75] (Table 1). Anti-HEV IgM and HEV RNA were detected in serum and fecal samples, thus providing evidence of recent exposure. All HEV strains clustered within the subtype HEV-3c, which is frequently found among Brazilian swine herds. The main risk factors associated with exposure to HEV were homelessness, crack-cocaine use ≥40 months, and sharing crack-cocaine equipment [75].

### 3.5. Immunosuppression/Immunodeficiency and Chronic Liver Disease

HEV primarily causes acute infections resulting in self-limited hepatitis. However, persistent HEV (mostly HEV-3) infection with rapid progression to chronic hepatitis and cirrhosis has been demonstrated in immunocompromised patients. Solid organ transplant (SOT) recipients — who are usually immunosuppressed — [96], and patients with hematologic malignant diseases [97] are prone to becoming infected with HEV after organ transplant and blood transfusion. In Brazil, the anti-HEV prevalence in SOT recipients seems to not be different from control groups. Some cross-sectional studies have assessed HEV RNA and antibodies in SOT recipients [10,67,68,69,70]. Commonly, in these studies, one of two (rarely both), HEV RNA or anti-HEV, were detected in the kidney and liver transplant recipients [10,70].

Anti-HEV IgG prevalence rates in HIV-seropositive selected groups ranged from 4.1% to 10.7% [11,76,77], not significantly differing from those reported in blood donors from the same Brazilian regions (Table 1). HEV RNA positivity, indicating ongoing infection, seems to not be frequent nor coincident with anti-HEV IgM or IgG detection in these cross-sectional studies [11,12]. For example, in a cohort of 360 HIV-seropositive patients from Rio Grande do Sul, South Brazil, eight had HEV RNA detected in the absence of anti-HEV IgM. However, neither the subsequent serum samples showed anti-HEV IgM or IgG seroconversion nor persistent viremia, thus excluding chronic infection, as pointed out by Moss da Silva et al. [11].

Patients with chronic liver disease have an increased risk of evolving into a more severe outcome, particularly to acute-on-chronic liver failure (ACLF), when superinfected with HEV [98,99,100]. In India, which is considered hyperendemic for HEV infection, Kumar Acharya et al. (84) prospectively evaluated 107 patients with cirrhosis—most of them caused by viral hepatitis (HBV and HEV) and continuous alcohol consumption—and compared them with a control group. The authors concluded that CLD patients superinfected with HEV can develop rapid cirrhosis decompensation due to ACLF. In Brazil, no prospective study has been conducted to evaluate HEV infection outcomes in patients with pre-existing liver diseases. Bricks et al., 2015–2016 developed a cross-sectional study and reported a 10.2% (95% CI 8.0% to 12.8%) anti-HEV prevalence among 618 patients chronically infected with hepatitis C virus (HCV) from São Paulo state, Southeast Brazil [72].

### 3.6. General Population and Blood Donors

Systematic reviews and meta-analyses of studies from North and South American countries have shown highly heterogeneous prevalence data. Sensitivity and specificity nuances between the EIA-based assays could partly explain such heterogeneity observed, particularly among HEV seroprevalence reported data from Brazil. According to Villalobos et al. [101], if only the studies using one of the two Abbott or Wantai assays were considered, either a 4% (95% CI: 2%–7%) or a 15% (95% CI: 11%–21%) polled anti-HEV prevalence would be estimated. Indeed, the earliest studies in Brazil used either the Biokit, (Barcelona, Spain) or the Abbott (Chicago, IL, USA) assays (Table 1) since the Wantai’s was not commercially available before 2013. Since then, the RecomWell HEV IgG/IgM (Mikrogen, Neuried, Germany) or the WANTAI HEV-IgG ELISA kit (Beijing, China) have been alternatively used. Nonetheless, an increasing anti-HEV seroprevalence independent of the assay has been observed in Brazil, more precisely in the southern and southeastern regions. Therefore, whether the increased anti-HEV positivity can be attributed to the higher sensitivity of the Wantai’s relative to other assays, as suggested by some authors [102], is still controversial. According to our experience, both the Wantai and the Mikrogen assays yielded comparable anti-HEV positivity rates when applied to the same cohort (21.7% vs. 23.7%) [82].

A recent systematic review and meta-analysis, including 142 articles (the most from USA and Brazil) published from January 1994 to December 2016, estimated HEV seroprevalence in the Americas and revealed that the risk of HEV exposure was lower in Brazil and other South American countries than in the United States of America (USA) [103]. Overall, the main risk factors of HEV infection in the Americas were increasing age, contact with pigs or pig products, and poor socioeconomic conditions. However, determinants of HEV infection are not limited to socioeconomic standards, as demonstrated by Horvatits et al. [103] based on the results of their systematic review and meta-analysis. The authors pointed out that HEV seroprevalence is significantly higher in the USA than in Latin America, independent of the methodological quality, anti-HEV detection assay, cohort, or year of the study. Indeed, the USA had a higher estimated seroprevalence (9%, CI: 5–15.6%) than Brazil (4.2%, CI: 2.4%–7.1%; OR: 2.27 (1.25–4.13); *p* = 0.007) and the Mixed Caribbean (1%, OR: 8.33 (1.15–81.61); *p* = 0.04).

Nationwide HEV seroprevalence data in the general Brazilian population and blood donors are scarce and difficult to interpret. The earliest studies assessed selected groups with increased risk of HEV infection, such as immunocompromised/immunosuppressed individuals and chronic liver disease carriers. Due to the difficult access to remote areas of the Amazon rainforest, many researchers have assessed sample repositories of studies, which were not designed to evaluate HEV infection, but malaria and other Brazilian subtropical diseases. Because of the intrinsic relationship between poverty and the incidence of waterborne infectious diseases, it is important to note that distribution and access to clean water and essential sanitation services are geographically uneven between the five Brazilian regions. The North and Northeast remain below the national average regarding water supply, sanitation, and garbage collection; therefore, they have the highest incidence of acute diarrheal diseases, hepatitis A, and other viral gastrointestinal infections. Contrastingly, HEV prevalence has constantly shown an increasing trend from north to south of the country, independent of the cohort or study year (Table 1).

In the South of Brazil, Pandolfi et al. [80] reported a highly increased anti-HEV prevalence (40%) in blood donors from Passo Fundo, Rio Grande do Sul state, using an “in-house” indirect ELISA. Next, using the same validated method, the authors investigated the presence of anti-HEV IgG antibodies in 3000 serum samples obtained from the general population of Passo Fundo, Caxias do Sul and Santa Maria between April and May 2019. The highest anti-HEV positivity was found in Passo Fundo (65.5%), followed by Caxias do Sul (57.4%) and Santa Maria (54.4%) citizens. The authors could not point out a specific risk factor distinguishing the studied population from other subsets from the same region [81]. For example, the consumption of pork and pork-derived food stuff (not rarely undercooked) is not an exclusive characteristic of the Rio Grande do Sul (RS) population but also a food custom of the Paraná (PR) and Santa Catarina (SC) states. Previous studies developed between 2013 and 2018 reported anti-HEV prevalence of 18.7% [70], 10% [79], and 7.1% [11] in blood donors from Porto Alegre (RS), Itajaí Valley (SC), and Rio Grande (RS) municipalities, respectively. Hardtke et al. [78] reported similarly high anti-HEV prevalence in pregnant women (19%) and female blood donors from Curitiba (PR), using cryopreserved samples obtained in 2002–2003. These data contrast with the low prevalence in blood donors from the Northeast (0.9%) [84] and North (0.4%) [71] (Table 1), demonstrating a paradigm shift of the binomial poverty−higher risk of HEV infection in developing countries. As a matter of fact, the estimated anti-HEV prevalence in the USA, a high-income country with a 0.92 HDI (human development index), was significantly higher than in Brazil (HDI 0.76) and other poorer South American countries with suboptimal hygiene and sanitation conditions [103].

Although most commercial swine farming and pork-derivative food processing in Brazil are centralized in the three southern states, consuming raw or undercooked meat or pork liver was not significantly associated with higher HEV prevalence; the only factor significantly associated with the outcome was living in a rural area, as shown for blood donors from Rio Grande municipality [11]. Nevertheless, the dietary preference for pork meat and pork-derivative food-stuff is not the only source of infection, as vegetables and fruits can be contaminated by pork manure spreading on soil (as a fertilizer) [81]. Nevertheless, only a few cross-sectional studies have addressed questions regarding HEV foodborne transmission since most of them analyzed samples from their biorepositories or provided by other research groups into the scope of unrelated projects.

Also, in the southeastern region, an unusually high HEV seroprevalence (up to 24%) was found in a sampled population of a small rural town of São Paulo [82]. In parallel with the increased HEV prevalence, our study revealed a declining prevalence of hepatitis A in the studied population. It is important to emphasize that the water chlorination and fluoridation implemented in the mid 1970’s and sewage treatment in the late 1980s coincided with the sharp decrease in HAV seroprevalence among individuals born in these two-time intervals. On the other hand, anti-HEV positivity had no association with low socioeconomic status and education. In the 1990s, many Brazilian swine farmers imported wild boars from Europe and Canada to breed them with domestic pigs, resulting in a fattest pig, the so-called “javaporco (boar-pig).” A widespread (intentional or unintentional) release of half-bred feral pigs followed, and São Paulo was the most affected state. Therefore, wild boars are the only animal species whose hunting has been authorized in Brazil for population control since 2013 [82]. Although hunting nor the consumption of game meat were addressed, the hypothesis of HEV-3 zoonotic transmission should not be discarded in the studied population.

In the general population and blood donors, increasing age was the only risk factor significantly associated with higher anti-HEV prevalence. Although HEV-3 is the only genotype identified in Brazil, with high homology between human and swine isolates, handling pigs was not relevant regarding HEV exposure, even within rural population cohorts. However, being born in southern or southeastern regions was significantly associated (*p* < 0.001) with a higher risk of exposure to HEV in blood donors from Central Brazil [83]. Similarly, HEV seroprevalence in rural workers settled in Acre and Rondonia (northern region)—most of them migrants from the southern and southeastern regions—has been substantially higher (twice to four times) than in the native (rural or riverine population (Table 1). Further studies addressing lifestyle, animal reservoirs and human relationships, environmental surveillance, and climatic changes are needed to evaluate the HEV epidemiology dynamics in Brazil.

## 4. Experimental HEV Infection Using Animal Models

Historically, the first attempt to reproduce HEV infection from one to another host was achieved by Balayan et al., 1983, who described the transmission of the putative etiological agent of acute non-A, non-B hepatitis via the fecal-oral route to a volunteer immune to HAV infection [104]. The inoculum consisted of pooled fecal suspensions obtained from nine subjects affected by an outbreak of acute hepatitis in a Soviet military camp in Afghanistan. Following, the intravenous inoculation of the virus-containing stool extract of the volunteer reproduced acute hepatitis in cynomolgus monkeys [104], whose infectious bile was used to construct recombinant complementary HEV DNA (cDNA) libraries [105], and further in vivo experiments confirmed the susceptibility of lambs inoculated with the fecal suspension of a patient with hepatitis E, as well as the passage in lambs of a piglet-derived HEV [106]. In Brazil, our research group successfully reproduced the HEV genotype 3 (HEV-3) infection in cynomolgus using virus-containing human fecal suspensions as inoculum [107,108,109,110]. The serological and molecular evidence of swine [13,41] and cynomolgus naturally infected with HEV-3 [109] confirmed, for the first time in South America, the potential zoonotic transmission of HEV-3 circulating among domestic pigs and other animal species. A previous study demonstrated the natural and experimental transmission of a European wild boar-derived HEV-3 to domestic pigs [111], raising awareness to the public health and livestock production’s safety.

In addition to macaques and pigs, other animals such as ferrets, humanized and non-humanized mice, rats, and Mongolian gerbils have been considered models for HEV infection [112,113]. The natural susceptibility of farmed rabbits to HEV was demonstrated by Lhomme et al. [114], and other authors have confirmed that specific pathogen-free (SPF) rabbits were susceptible to infection with rabbit HEV-3 (CHN-BJ-rb14). However, the human HEV-3 (JRC-HE3) recovered from primary infected rabbits was not transmissible to naïve rabbits [115]. On the other hand, a swine HEV-4 strain replicated and induced histological injury in the ovarian tissues of rabbits experimentally infected. Histological changes in rabbit ovarian tissue included scattered cell necrosis, lymphocyte infiltration, and accelerated apoptosis [116]. Recently, He and colleagues (2022) established an immunocompromised rabbit model to study chronic hepatitis E and to evaluate the efficacy of a Chinese vaccine and antivirals [117].

Experimental infection using animal models and environment surveillance studies are scarce in Brazil and other South American countries, mainly due to the high cost and limited financial support dedicated to research projects in these research fields. Furthermore, contrasting with the considerable increase in HAV outbreaks affecting young adults (18−39 years old) [118], there have been few sporadic cases of HEV infection, mainly in immunocompromised persons [118]. In 2014, we compared the virulence of HEV-3 recovered from swine feces and viremic patient sera. The results of this study confirmed the feasibility of the *Macaca fascicularis* model that reproduced a subclinical HEV infection with the detection of HEV RNA between the fifth and fifty-third day post inoculation (dpi) [107]. Animals presented a mild inflammation of liver tissues and discrete elevation of liver enzymes. Seroconversion to anti-HEV IgM and/or IgG was detected in seven of the eight infected monkeys, and anti-HEV IgA in the salivary samples of three animals. Interestingly, all infected monkeys showed severe lymphopenia and a trend toward monocytosis, which coincided with the elevation in alanine aminotransferase and antibody titers. The hypothesized HEV spillover skill was confirmed for either HEV-3 recovered from Brazilian and Dutch swine, or for the Argentinean human HEV-3 strains.

The broad circulation of the zoonotic HEV in South America motivated our group to design a new study to evaluate the hypothesis of inducing a persistent HEV-3 in cynomolgus monkeys treated with the immunosuppressant tacrolimus, aiming to develop an NHP model for studying the pathogenesis of chronic hepatitis E [108]. HEV replication and prolonged active lymph-histiocytic reactivity were detected in the liver parenchyma at 69 dpi [108]. Animals showed moderate weight loss, alopecia, and herpes virus opportunistic infection due to their pre- and post-inoculation immunosuppressive conditions. In our study, the tacrolimus-induced chronic hepatitis E was characterized by a mild increase in liver enzyme levels, persistent HEV RNA presence in blood and liver, and fecal samples within three months of infection, according to recent criteria to classify chronic hepatitis E [119]. In our study, three out of the four immunosuppressed monkeys showed evident hepatocellular ballooning degeneration, mild to severe macro- and microvesicular steatosis (zone 1), scattered hepatocellular apoptosis, and lobular focal inflammation. At 69 dpi, liver biopsies of all infected monkeys revealed evident hepatocyte ballooning degeneration (zone 3), discrete hepatocellular apoptosis and, at most, a mild portal and intra-acinar focal inflammation. At 160 dpi, the three chronically HEV-3-infected monkeys showed microscopic features (piecemeal necrosis) corresponding to chronic hepatitis in the absence of fibrosis and cirrhosis in liver parenchyma, as shown in tacrolimus-immunosuppressed solid organ transplant (SOT) recipients. The cause-effect relationship between HEV infection and tacrolimus treatment was confirmed in this experiment [108].

## 5. Hepatitis E Virus in Brazil: Forth and Beyond

In contrast to hepatitis A virus (HAV)—also an enterically transmitted virus—HEV infection does not seem to be associated with low-income and suboptimal hygiene and sanitation infrastructure. Considering the unequal socioeconomic patterns observed between the five Brazilian geographic regions, the earliest studies on HEV epidemiology have focused on impoverished selected populations regardless of the geographic region. No one has found a significant increased seroprevalence associated with low-income condition. Regarding hepatitis A endemicity, anti-HAV seroprevalence is directly associated with socio-economic variable levels, such as water supply, regular water supply, and sewage disposal, as reported by Pereira et al. [120]. The authors analyzed data from a population-based study from 2005 to 2009 to evaluate the predictive factors for hepatitis A in the North, South, and Southeast regions.

Although anti-HAV prevalence has been estimated only for children and adolescents (5–19 years old), the authors pointed out that HAV endemicity has decreased from high to intermediate in the North, and to a low level in the South and Southeast—probably reflecting the socio-economic improvement achieved after the implementation of the Growth Acceleration Program by the Brazilian government, in 2007. Nevertheless, HAV infection is still highly endemic in the North region, with anti-HAV prevalence rates up 70% in children and adolescents, much higher than those found in South and Southeast regions, of approximately 40% [120]. The opposite has been observed concerning HEV seroprevalence, which has substantially increased in the last decade, notably in the South and Southeast—an inverse correlation with the higher human development index (0.754 and 0.766) relative to the North and Northeast (0.667 and 0.663) [121].

HEV-3 is highly prevalent in the Brazilian swine herd, on large-scale or family farms. As HEV-3 is the sole genotype identified in Brazilian HEV-infected patients and the infection is clinically silent, few cases of acute hepatitis E have been reported in Brazil. All human isolates showed a close relationship with the porcine isolates, evidencing zoonotic transmission. The possibility of spillover to other animal species must be surveilled.

The results of our experiment in cynomolgus monkeys evidenced a persistent HEV infection associated with tacrolimus-induced immunosuppression, thus reinforcing the potential risk of chronic hepatitis E for immunocompromised individuals, such as SOT recipients. Considering that the detection of HEV RNA in immunocompetent individuals, such as blood donors, is not uncommon [122], organ donor screening for HEV RNA and antigen has been considered in some countries, as a public health measure to prevent donor-to-recipient transmission, especially among immunosuppressed SOT recipients [93,123]. In addition, immunosuppressed patients should avoid consuming undercooked pork meat, since HEV-3 infections can progress to chronic hepatitis with life-threatening consequences.

Finally, HEV and its zoonotic potential are still a subject of great interest in the One Health approach. Studies focusing on the description of HEV genotypes/subtypes characterization in animal, human, and environmental origins are increasing, as well as new animal reservoir identification. Importantly, different geographical areas might consider regional social and cultural habits. Brazil is third in the international market for bovine, swine, and poultry livestock [124]; donkey meat has also been commercialized for Brazilian customers or exports [125]. Considering the potential risk of foodborne transmission, HEV detection and characterization methods applied to unprocessed animal derivatives destined for internal demand and exportation should be mandatory. Likewise, researchers from the area should consider environmental samples to evaluate the impact of intensive animal production. In parallel, surveillance of suspect hepatitis E acute hepatitis in humans could clarify the real impact of zoonotic hepatitis E in Brazil.

## Figures and Tables

**Table 1 viruses-15-00548-t001:** Hepatitis E virus seroprevalence in Brazil.

Selected Population Subsets								
Region	UF	Sampling Location	Year	Cohort	Sample Size	Prevalence (%)	Anti-HEV Assay	Reference
Midwest	MT	Brazilian Amazon	1994	AVH outbreak	97	6.1	Not informed	[52]
Midwest	MT	Brazilian Amazon	1997	AVH (nonA-C)	82	12	Not informed	[51]
Northeast	BA	Salvador	1994–1996	AVH (nonA-C)	17	17.7	Not informed	[55]
				AVH (HAV)	24	29		
				Hd	392	0		
				CLD/Sch	30	11		
				BD	200	2		
Southeast	RJ	Rio de Janeiro	1994–1996	BD	93	4.3	ABBOTT^1^	[49]
				CRD/Hd	65	6.2		
				IDU	102	11.8		
				Pregnant women	304	1.0		
				Rural	145	2.1		
Rural populations								
Midwest	MT	Swine farms	2009–2010	Rural	310	8.4	MP Diagnostics	[56]
North	AC	Western Brazilian Amazon	2004	Rural settlers	425	12.9	MIKROGEN ^2^	[57]
North	RO	Western Brazilian Amazon	2013	Rural settlers Rural Urban	281 73 47	14.6 8.2 4.2	BIOKIT^3^	[58]
Midwest	GO MS	Central Brazil	2011	Rural settlers	923	3.9	MIKROGEN ^2^	[59]
Midwest	GO	Central Brazil	2011	Rural settlers.	464	3.4	MIKROGEN ^2^	[60]
Brazilian Traditional Peoples								
North	AC AM	Western Brazilian Amazon	1997	Riverine	349	4.0	ABBOTT ^1^	[54]
North	PA	Eastern Brazilian Amazon	2012	Riverine	172	4.1	BIOKIT ^3^	[61]
North	PA	Eastern Brazilian Amazon	2015	Afro-descendants	535	1.6	MIKROGEN ^2^	[62]
North	TO	Apinajé villages	--	Indigenous Urban	506 175	0.19 0	MIKROGEN ^2^	[63]
North	AM	Yanomami villages	2015	Indigenous	430	2.8–5.7	MIKROGEN ^2^	[64]
Selected population groups according to risk category								
North	PA	Eastern Brazilian Amazon	1993–2014	AVH (nonA-C)	318	5.9	MIKROGEN ^2^	[65]
Midwest	GO	Central Brazil	2012–2014	AVH (nonA-C)	379	5.3	MIKROGEN ^2^	[66]
Southeast	SP	São Paulo	1998–2007	SOT	96	3.1	MIKROGEN ^2^	[67]
Southeast	SP	São Paulo	2001–2011	SOT	192	15.0	MIKROGEN ^2^	[68]
Midwest	GO	Goiania	2014	SOT	316	2.5	MIKROGEN ^2^	[69]
South	RS	Porto Alegre	2013–2018	SOT	80	18.7	WANTAI ^4^	[70]
South	RS	Porto Alegre	2013–2018	CLD	80	22.5	WANTAI ^4^	[70]
North	AM	Manaus	2002	CRD/Hd	192	0.5	ABBOTT ^1^	[71]
Southeast	SP	Piracicaba	2015–2016	CLD	618	10.2	WANTAI ^4^	[72]
Midwest	GO	Central Brasil	2010–2011	RWP	431	5.3	MIKROGEN ^2^	[73]
Midwest	MS	Central Brasil	2013–2015	CCU	698	14.2	WANTAI ^4^	[74]
North	PA	PA municipalities	2016–2018	CCU	437	16.7	MP Diag ^5^	[75]
South	RS	Porto Alegre	2013–2018	CCU	80	20.0	WANTAI ^4^	[70]
South	RS	Rio Grande	2012–2013	HIV	360	6.7	MIKROGEN ^2^	[11]
Southeast	SP	São Paulo state	2007–2013	HIV	354	10.7	MIKROGEN ^2^	[76]
Northeast	PE	Recife	2016–2017	HIV	366	4.1	MIKROGEN ^2^	[77]
General population and blood donors								
Southeast	RJ	Rio de Janeiro	1996	GP (low-income, urban)	699	2.4	ABBOTT ^1^	[50]
Midwest	MT	Southern Brazilian Amazon	1997	GP (low-income, urban/rural)	299	3.3	ABBOTT ^1^	[53]
South	PR	Curitiba	2002–2003	BD (female) Pregnant women	199 209	26 19	WANTAI ^4^	[78]
South	SC	Itajaí Valley	2014	BD	300	10.0	WANTAI ^4^	[79]
South	RS	Rio Grande	2015	BD	281	7.1	MIKROGEN ^2^	[11]
South	RS	Passo Fundo	2015	BD	780	40.2	indirect ELISA ^5^	[80]
South	RS	Santa Maria Caxias do Sul Passo Fundo	2019	GP	1000 1000 1000	55.4 57.4 65.5	indirect ELISA ^5^	[81]
South	RS	Porto Alegre	2013–2018	BD	80	18.7	WANTAI ^4^	[70]
South	RS	Porto Alegre	2013–2018	GP	80	17.5	WANTAI ^4^	[70]
Southeast	SP	São Paulo	2014	BD	500	9.8	WANTAI ^4^	[79]
Southeast	SP	Cassia Coqueiros	2011–2013	GP	224	23.7 21.4	MIKROGEN ^2^ WANTAI ^4^	[82]
Midwest	MT	Cuiabá, and other municipalities	2009–2010	BD	110	4.0	MP Diagnostics	[56]
Midwest	MS	Campo Grande	2011	BD	250	6.4	WANTAI^4^	[83]
Northeast	PE	Recife	2021	BD	996	0.9	EUROIMMUN ^6^	[84]
North	AM	Manaus	2002	BD	184	0.4	ABBOTT ^1^	[71]
North	AC	Western Brazilian Amazon	2012–2013	GP	870	0.9	MIKROGEN ^2^	[85]

UF, Unidade Federativa (Federate Unit); AVH, acute hepatitis; hepatitis; SOT, solid organ transplant recipients; CLD, chronic liver disease; Schist, Schistosomiasis; CRD, chronic renal disease; Hd, hemodialysis patients; IDU, injecting drug users; CCU, crack-cocaine users; RWP, recyclable waste pickers; HIV, human immunodeficiency virus seropositive; BD, blood donors; GP, general population; **^1^** Abbott HEV EIA (Abbott Diagnostics Division, Chicago, IL, USA); **^2^** Recomwell HEV IgM/IgG, (Mikrogen, Neuried, Germany); **^3^** IgG/IgM_EIA (Biokit, Barcelona, Spain); **^4^** WANTAI HEV-IgG ELISA kit (Beijing, China); **^5^** indirect enzyme-linked immunosorbent assay [80]; **^6^** ANTI-HEV IgG EUROIMMUN (Lübeck, Germany)

## Data Availability

The datasets generated and analyzed during the current study are available in the GenBank repository under accession number A49F569.

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
