# Peer review of "Hepatitis E Virus Research in Brazil: Looking Back and Forwards"

_viruses, 2023, doi:10.3390/v15020548_

Round 1
Reviewer 1 Report
The authors summarized the prevalence of HEV infection in humans and animals of Brazil. They also compared the animal models of HEV. This is an interesting research but some issues need to be addressed before publication.
1. HEV-7 infection has also been reported in human beings. Please add this in Introduction section.
2. In line 37-39, the related references need to be added.
3. In the “Hepatitis E virus in animal reservoirs in Brazil” section, authors mainly discussed swine and wild boars. As far as I know, HEV strains have been detected in many other animals in Brazil (Moraes DFDSD, Mesquita JR, Dutra V, Nascimento MSJ. Systematic Review of Hepatitis E Virus in Brazil: A One-Health Approach of the Human-Animal-Environment Triad. Animals (Basel). 2021;11(8):2290). Please add these references in this section.
4. In the “HEV prevalence in the Brazilian population” section, authors listed the prevalence of anti-HEV in Table 1. Did the prevalence of HEV RNA also been detected in the references? It is suggested that the data of HEV RNA should be included.
5. In the “Experimental HEV infection using animal models” section, authors mainly discussed about non-human primates. Can authors compare the non-human primates with other animal models of HEV? Also, the potential use of animal models in further study of HEV should be discussed.
6. The title of the study is “Hepatitis E virus Research in Brazil: Background and Perspective”. However, this study mainly contained analysis of HEV seroprevalence in animal reservoirs and human beings. Can authors discuss more about the perspective of HEV study of Brazil?
7. Many typos are found in the manuscript. Proofreading is needed.
Author Response
Answers to the reviewers
Reviewer 1:
- We mention one report describing HEV-7 in humans (lines 36-37)
- References were added (lines 37-39 / 41-44)
- In the present Review manuscript, we aimed to review and discuss the Brazilian studies on HEV prevalence in animal hosts and human populations. “In the section Hepatitis E virus in animal reservoirs in Brazil”, the author focused on swine, which represents most of the studies in Brazil. Besides swine, only two studies reported HEV RNA detection in other animal hosts: Souza et al. (2018) reported the detection and characterisation of a novel HEV in wild rodents, and Bastos et al. (2021) detected the zoonotic HEV-3 in bovine liver samples. We mentioned both studies (lines 112-116 and 120-121). Before, our research group detected anti-HEV antibodies (not HEV RNA) in pigs, cows, dogs, chickens, and rodents (Vitral et al., 2005), as mentioned (lines 60-63 and 71-82).
- Table 1, entitled “Hepatitis E virus seroprevalence in Brazil”, depicts anti-HEV antibodies prevalence data. We mentioned HEV RNA data (not included in Table 1) throughout the manuscript. Only a few studies have reported HEV RNA detection in serum samples from solid organ transplant (SOT) recipients (lines 336-340) and a single case of acute hepatitis E (lines 171-177).
- As suggested by the reviewer, we mentioned other potential animal models for studying HEV infection (lines 542-547)
- In the present manuscript, the authors aimed to review and comment on the HEV prevalence data reported in Brazil in the last two decades. Most studies assessed the anti-HEV seroprevalence in human population subsets, whereas just a few assessed HEV RNA and phylogenetic analysis. HEV-3 is the sole genotype circulating in Brazil, and most infections are asymptomatic. So, we agree with the reviewer that the present review article misses “the perspective of HEV study in Brazil) Furthermore, we changed the Review article´s title to “Hepatitis E virus Research in Brazil: Looking back and forth”. Besides, we added a last paragraph in the Conclusion section, pointing out the lack of data on HEV surveillance in the environment and the distribution of genotypes/subtypes in animal hosts. (lines 587-599).
- Unintentional typographical errors were proofread.

Reviewer 2 Report
This paper is a review of HEV research in Brazil. This review covers HEV prevalence studies across different regions and populations in Brazil, genotypes and subtypes identified and animal research.
Please see the updated taxonomy for Hepeviridae and rewrite this review as necessary. After describing the taxonomy for Hepeviridae the authors should mention that their paper only covers the Paslahepevirus species in Brazil. https://www.microbiologyresearch.org/content/journal/jgv/10.1099/jgv.0.001778
Line 23. This genome length is specific for Paslahepevirus. The other species have their own characteristic genome lengths.
Line 26. The eight genotypes mentioned are from Paslahepevirus. The other species have their own sets of genotypes.
Line 37. East and South Asia, and Africa.
Lines 37-39. Add references for the results mentioned here.
Line 47. This is true except for Paslahepevirus genotypes 1 and 2 that only infect humans.
Line 55. Identification of animals infected with HEV using serology alone can be problematic as there is wide discordance between different serological assays with false positivity in some serological assays. See your own statement in line 367. This is hinted at in later sections of this review, but it could be useful to discuss the discordance problem with serologic assays both with animals and humans at this point in the review. cDNA sequencing and PCR detection are the gold standard.
Lines 73-76. Please delete mention of the Nepal study. It was retracted because the identified HEV sequences was discovered to be a laboratory contamination from an isolate from another country.
Line 78. HEV genotype and subtype identification can only be done through cDNA sequencing and PCR. The preceding section of this paper only mentioned serological assays and some readers may believe that genotype and subtype identification was also done through serological assays.
Line 122. Replace inner with Brazilian.
Line 143 and Table 1. Given that at least five different serological assays were used and given the discordance between serological assays, direct comparisons between populations tested with different assays cannot be made. It could be useful to describe the serology discordance issue here.
Line 163. It might be better to use related instead of retrieved here.
Line 171. At the end of this line 0,3% should be 0.3%.
Line 173. Outbreak should be outbreaks.
Line 213. Were should be was.
Line 228. Replace associated the HEV with associated with HEV.
Line 340. Delete acute on.
Line 358. Asia and Africa.
Line 420. Should costume be custom?
Author Response
Reviewer 2:
- Hepeviridae taxonomy text was rewritten, accordingly. (lines 23-37)
- East and South Asia, and Africa. (line 42)
- In Brazil, as in other countries in South America, HEV genotype 3 (HEV-3) is, currently, the only genotype identified in humans (5–8), animal hosts (9–18), and environmental matrices (19–21). References added.
- We rewrote, as recommended by the reviewer: “Except for Paslahepevirus genotypes 1 and 2, it is known that the hepatitis E virus (HEV), among other relevant viruses that cause hepatitis, is well characterized as a zoonotic virus... (Lines 51-56)
- We agree with the reviewer that HEV RNA detection by PCR and cDNA sequencing are the gold standard for HEV identification in animals. In the second paragraph of the “Hepatitis E virus in animal in animal reservoirs in Brazil”, we meant to set a timeline of the HEV research in Brazil. The earliest study provided serological evidence of HEV circulating among several animal species (Vitral et al., 2005) in Rio de Janeiro swine farms and corroborated previous serological studies. Considering that swine have emerged as a potential zoonotic host in USA and other countries, our research group focused on HEV detection (by PCR and cDNA sequencing) in farmed pigs from the Southeast and Midwest (Santos et al., 2011), thus confirming the circulation of HEV genotype 3 in domestic and farmed pigs in Brazil. We rewrote the paragraph, accordingly (Lines 60-68 and 73-75).
- We deleted the retracted study from Nepal, which have reported the detection of HEV-3 in rats as advised by the reviewer.
- We rewrote the paragraph to emphasize that the high anti-HEV positivity reported in 1990-2000 studies contrasts with further studies, which employed different enzyme immunoassays, most likely due to the improved specificity of the next-generation assays. (Lines 147-151).
- We meant “acute-on-chronic liver failure”.
- We replaced to “East and South Asia, and Africa” all mentions to HEV genotypes 1 and 2 prevalence.
- We replaced “costume” to “food custom”.

Round 2
Reviewer 1 Report
The authors have revised the manuscript accordingly. However, some issues need to be addressed before publication can be considered.
1. Page 2, line 55: As far as I know, the main animal hosts of HEV include swine and rabbits. It is suggested that authors should cited the references of rabbit HEV here.
2. Page 2, line 61-62: The authors mentioned “anti-HEV antibodies”. Anti-HEV antibodies should include anti-HEV IgM and IgG. Did authors analyze the data of anti-HEV IgM? Also, “anti-HEV IgG antibodies” should be replaced by “anti-HEV IgG antibody”.
3. Page 2, line 67: It is interesting that the anti-HEV positive rats which were exposed to swine HEV3 may be infected by HEV3. Recently, rat HEV has been detected (PMID: 35364118; PMID: 34718428) and zoonotic infection has also been reported. Did authors consider the possibility of rat HEV infection? It would be better to cite the references and discuss about it.
4. Page 3, line 128: Authors mentioned seroprevalence of pregnant women here, but no they did not further discuss about it. HEV can cause high mortality in pregnant women.Did authors collect the data of HEV infection in pregnant women?
5. Page 12, line 491: Authors claimed that these animal models cannot be infected by human-derived HEV-3. However, some literatures showed that rabbit model can be infected by human HEV-3 (PMID: 31339458; PMID: 35278241). Thus, it needs to be revised.
Author Response
Dear Reviewer,
We are thankful for the valorous comments and suggestions. Please, find our corrections in the manuscript, highlighted in green.
- Page 2, line 55: As far as I know, the main animal hosts of HEV include swine and rabbits. It is suggested that authors should cited the references of rabbit HEV here.
Citations added (line 55 and References)
- Page 2, line 61-62: The authors mentioned “anti-HEV antibodies”. Anti-HEV antibodies should include anti-HEV IgM and IgG. Did authors analyze the data of anti-HEV IgM? Also, “anti-HEV IgG antibodies” should be replaced by “anti-HEV IgG antibody”.
We corrected it to “anti-HEV IgG antibodies”. We kept the plural (antibodies) because we meant several species-specific IgG antibodies (lines 62-63).
- Page 2, line 67: It is interesting that the anti-HEV positive rats which were exposed to swine HEV3 may be infected by HEV3. Recently, rat HEV has been detected (PMID: 35364118; PMID: 34718428) and zoonotic infection has also been reported. Did authors consider the possibility of rat HEV infection? It would be better to cite the references and discuss about it. DEBORA, você gostaria de discutir?
We added the reference the Reviewer recommended (lines 71-73).
- Page 3, line 128: Authors mentioned seroprevalence of pregnant women here, but no they did not further discuss about it. HEV can cause high mortality in pregnant women. Did authors collect the data of HEV infection in pregnant women?
Trinta et al. (2001) found a 1% anti-HEV seroprevalence in pregnant women, which was lower than in blood donors from the same city, as shown in Table 1 (a). We included the pregnant data in the table.
- Page 12, line 491: Authors claimed that these animal models cannot be infected by human-derived HEV-3. However, some literatures showed that rabbit model can be infected by human HEV-3 (PMID: 31339458; PMID: 35278241). Thus, it needs to be revised.
We mentioned studies of experimental infection of SPF rabbits with HEV-3, and an immunosuppressed rabbit model to studying chronic hepatitis E and to evaluate the efficacy of a Chinese vaccine (lines 495-504 and References).

Reviewer 2 Report
The authors have answered most of my queries. There are a few additional edits that should be made.
Lines 46-47. Replace the highlighted region with primarily on animals infected with P. balayani and distinct human populations.
Line 50. It would be better to replace Paslahepevirus with P. balayani.
Line 66. Please be careful here. Serological positivity in rats may not have been exposed to virus from Paslahepevirus as most serological assays cross-react between Paslahepevirus and Rocahepevirus.
Line 66. Replace have with had.
Line 72. Delete then.
Lines 110-111. These two animals were infected with HEV from the Rocahepevirus genus. This should be pointed out here as earlier you said this review would primarily look at Paslahepevirus hosts.
Line 323. Replace not differ with no different.
Line 377. Wantai has high sensitivity and specificity. Low specificity means there should be an increased number of false positives.
Line 487. Macaques can be infected with both genotype 1 and 3.
Author Response
Reviewer 2
The authors have answered most of my queries. There are a few additional edits that should be made.
Lines 46-47. Replace the highlighted region with primarily on animals infected with P. balayani and distinct human populations.
JMO: Suggestion accepted, and text highlighted in green (Line 47)
Line 50. It would be better to replace Paslahepevirus with P. balayani.
JMO: Suggestion accepted, and text highlighted in green (Line 51)
Line 66. Please be careful here. Serological positivity in rats may not have been exposed to virus from Paslahepevirus as most serological assays cross-react between Paslahepevirus and Rocahepevirus.
JMO: Suggestion accepted, and the mention “it is speculative that the two anti-HEV positive rats have been exposed to the swine HEV genotype 3” was deleted. The corrected text is highlighted in green (Lines 65-67; 71-74)
Line 66. Replace have with had.
JMO: The text “it is speculative that the two anti-HEV positive rats had been exposed to the swine HEV genotype 3” was deleted.
Line 72. Delete then.
JMO: Suggestion accepted, and text highlighted in green (Line 74)
Lines 110-111. These two animals were infected with HEV from the Rocahepevirus genus. This should be pointed out here as earlier you said this review would primarily look at Paslahepevirus hosts.
DRLS: We pointed out that the mentioned …” identified a novel orthohepevirus in blood samples from Necromys lasiurus and Calomys tener species, bringing perspective to host range and viral diversity of the Hepeviridae family in Brazil, including the genus Rocahepevirus”(lines 113-115).
Line 323. Replace not differ with no different.
JMO: Suggestion accepted, and text highlighted in green (Lines 327)
Line 377. Wantai has high sensitivity and specificity. Low specificity means there should be an increased number of false positives.
JMO: Suggestion accepted, and text highlighted in green (Line 380-381)
Line 487. Macaques can be infected with both genotype 1 and 3.
MAP: We mentioned studies of experimental infection of SPF rabbits with HEV-3, and an immunosuppressed rabbit model to studying chronic hepatitis E (Lines 496-505).
